# Successful Removal of *Angiostrongylus cantonensis* Larvae from the Central Nervous System of Rats 7- and 14-Days Post-Infection Using a Product Containing Moxidectin, Sarolaner and Pyrantel Embonate (Simparica Trio™) in Experimental Infections

**DOI:** 10.3390/pathogens12020305

**Published:** 2023-02-12

**Authors:** Michaela Henry, Richard Malik, Jan Šlapeta, Rogan Lee

**Affiliations:** 1Sydney School of Veterinary Science, Faculty of Science, The University of Sydney, Sydney, NSW 2006, Australia; 2Centre for Veterinary Education, The University of Sydney, Sydney, NSW 2006, Australia; 3The University of Sydney Institute for Infectious Diseases, Westmead Hospital, The University of Sydney, Sydney, NSW 2006, Australia

**Keywords:** *Angiostrongylus cantonensis*, rat lungworm, moxidectin, Simparica

## Abstract

*Angiostrongylus cantonensis* is a nematode with an indirect lifecycle, using molluscs as intermediate hosts. Rats are the definitive host. By administering a suitable anthelmintic, at an appropriate interval, the risk of clinical neuroangiostrongyliasis occurring in paratenic hosts (e.g., dogs, man) can be eliminated. We wanted to determine if infective larvae (L3) of *A. cantonensis* can be safely killed during their migration through the central nervous system (CNS) by oral administration of an anthelmintic combination containing moxidectin (480 µg/kg, Simparica Trio™; M-S-P), thereby preventing patent infections in rats. Eighteen rats were used: ten received oral M-S-P every four weeks; eight rats were used as controls. Rats were initially given M-S-P as a chew to eat, but an acquired food aversion meant that subsequent doses were given by orogastric lavage. All 18 rats were challenged once or twice with approximately 30 L3 *A. cantonensis* larvae via orogastric lavage. Infection status was determined by faecal analysis using the Baermann technique and necropsy examination of the heart, pulmonary arteries and lungs. Eight out of ten rats dosed with M-S-P had zero lungworms at necropsy; a single female worm was detected in each of the remaining two rats. No treated rats had L1 larvae in faeces. In contrast, all eight controls were infected with patent infections, with a median of 14.5 worms per rat detected at necropsy. The difference in infection rates was significant (two tailed Fishers Exact; *p* = 0.0011). Moxidectin given orally once every month killed migrating larvae before they reached the pulmonary arteries in 80% of treated rats, while in 20%, only a single female worm was present. Considering the short half-life of moxidectin in the rat, it is likely that the effectiveness of moxidectin is due to larvicidal action on migrating L3, L4 and L5 larvae in the brain parenchyma or subarachnoid space, either 7 days (L3/L4 in cerebrum and spinal cord) or 14 days (L4/L5 in cerebrum and subarachnoid space) after inoculation. This study is a prelude for future research to determine if monthly moxidectin administration orally as M-S-P could prevent symptomatic neuroangiostrongyliasis in dogs.

## 1. Introduction

The rat lungworm *Angiostrongylus cantonensis* is a nematode with a complex indirect life cycle. This involves an unusual obligatory migration stage, in which L3 larvae migrate through the central nervous system (CNS), where they grow, moult and further mature. The parasite uses molluscs, such as snails and slugs, as intermediate hosts, while rats are the definitive host [1]. Rats and paratenic hosts become infected by ingesting L3 larvae present in tissues of the intermediate and paratenic (transport) hosts. *A. cantonensis* usually does not produce overt clinical signs in rats, unless the definitive host ingests a heavy dose of infective larvae over a short time. If this happens, verminous pneumonia develops, clinically manifested by dyspnoea and reduced exercise tolerance [1,2,3,4]. Rarely, transient neurological signs occur; although, usually despite extensive neural migration by larvae, neurological signs in rats are absent or unappreciated.

Adult lungworms live in the pulmonary arteries of several rat species. The life cycle is complex, involving snails as intermediate hosts, and a variety of different species as paratenic hosts. Male and female lungworm mate in the pulmonary arteries. Females lay eggs which embolise to the pulmonary parenchyma. After hatching, first stage L1 larvae move up the mucociliary escalator, are swallowed, and appear in faecal pellets of the rats, which are attractive to snails. Snails become infected by ingestion of infective first stage larvae, which develop and mature into L3 larvae. The life cycle is completed when rats ingest snails. L3 larvae penetrate the intestine of the rat, travel in the portal circulation to the liver, and then rapidly reach the systematic circulation. The most unusual aspect of the lifecycle involves an obligatory migration of L3 larvae in the rat central nervous system (CNS). Larvae migrate widely through the CNS, including the spinal cord, brain and optic nerves, growing, moulting and growing further until they are ready to leave the CNS via the arachnoid villi to make their way back to the right ventricle and pulmonary arteries [1,2,3,4].

In contradistinction to the situation in rats, the migration of infective L3 larvae can cause serious disease in accidental hosts such as dogs, wildlife (birds, possums, bats) and humans [1,2,3,4,5]. Rat lungworm disease (neuroangiostrongyliasis) is most often manifested as eosinophilic meningoencephalitis [3,4], and peripheral eosinophilia is often reported in canine and human dead-end hosts [2]. The actual migration of larvae through the CNS and the inflammatory response thereby incited can together give rise to hindlimb weakness, paralysis and even death, if the animal does not receive treatment. This is due to severe damage to the CNS, especially the spinal cord and cauda equina, and in some cases the brain [4,5,6].

To minimise the prevalence of clinical rat lungworm disease, it is necessary to interrupt critical portions of the life cycle of *A. cantonensis*, thereby limiting the number of infections, and their extent, in dogs, wildlife and man [6,7,8,9]. Although it is certainly helpful to reduce the number of rats and snails in the environment, the strategy most likely to be successful at preventing canine infections is the administration of prophylactic anthelmintics. The key consideration for anthelmintic choice is the half-life of the drug and its dosing frequency.

The macrocyclic lactone moxidectin has a long half-life in most species, including dogs. When given at monthly intervals, it has been shown to be highly effective in preventing infection of dogs with *Angiostrongylus vasorum* [10,11], a closely related parasite of dogs which has a similar lifecycle, but without the novel larval migration though the CNS. It is therefore to be expected that moxidectin, if given at the same dosage interval, is also likely to prevent infective larvae of *A. cantonensis* reaching and damaging the CNS of dogs [12,13]. This is because L3 larvae migrating through canine tissues on their way to the CNS will be killed in transit if moxidectin concentrations are sufficiently high in the blood and extracellular fluid, which is likely for up to a month after dosing with systemic moxidectin, as a result of its long half-life [14,15].

The aim of this study was to evaluate the efficacy of an orally administered combination of sarolaner/moxidectin/pyrantel embonate (Simparica Trio™ [M-S-P]; Zoetis; [16]) given every four weeks against subsequent challenge with *A. cantonensis* L3 larvae in rats. When we designed this experiment, the pharmacokinetics of moxidectin in the rat had not yet been determined, and we erroneously thought that the half-life in the rat would be comparable to that in the dog (see later).

## 2. Materials and Methods

### 2.1. Animals

Eighteen juvenile male rats (*Rattus norvegicus*; Wistar strain) were used for this study. All rats were approx. 4–5 months-of-age at the start of the study; Group 1 rats were 19 weeks, Group 2 rats were 14 weeks, while Group 3 and 4 rats were 11 weeks old at the start of the experiment. The median weight of the rats at the start of the experiment was 505 g. Rats were housed in pairs in cages with an appropriate substrate (wood shavings), enrichment items (e.g., cardboard rolls), and were fed commercial rat cubes and provided with fresh tap water ad libitum. 

The first dose of M-S-P was given to the rats in tablet form. Our assumption was that a product developed to be palatable chew for dogs, would also be palatable for rats, and a pilot dose given to rats (not part of this study cohort) was consumed. Food was withheld from each treatment rat overnight to ensure the subjects were hungry. Each rat was isolated in their own cage the following day with no bedding and just the half tablet, with rats monitored over the course of the day. This proved satisfactory on the first occasion that the product was provided. However, when given the second and third doses of M-S-P, rats were reluctant to eat the ‘chew’ voluntarily, probably because they had developed a taste aversion. To circumvent this, the half tablet was dissolved in 3 mL of distilled water to form a suspension and was administered to each rat via orogastric lavage under light isoflurane anaesthesia. Rats were monitored shortly after the procedure for any evidence of aspiration, regurgitation or vomiting. 

All 18 rats were each challenged, on one or two occasions, with approx. 30 *A. cantonensis* L3 larvae. Infective L3 larvae were first freshly harvested from macerated tissues of chronically infected snails 2–3 h prior to administration. Larvae were administered to rats via oral gavage using a plastic pipette under light isoflurane anaesthesia. One of the investigators (RL) counted out 30 larvae as they were sucked up into a pipette tip; the approximate final volume was made up with distilled water to a volume of 500 µL. These larvae were then instilled into the distal oesophagus, although it was not possible to control for the loss of some larvae (via subsequent regurgitation and vomiting) during the gavage process. We expected that 50–75% of administered larvae would reach the stomach due to losses of larvae during this procedure. 

The control group consisted of eight rats, four of which were given 30 L3 at 2 weeks (Group 1), and four of which were given the same dose at 7 weeks (Group 2). The treatment group consisted of ten rats, divided into two groups of five, that had been given M-S-P orally at zero weeks, four weeks and eight weeks by voluntary intake (t = 0), and subsequently using gastric lavage (t = 4 and 8 weeks) (Table 1). The monthly administration of M-S-P was chosen to mimic the situation whereby pet dogs are given this product on an ongoing monthly basis, while potentially being exposed to rat lungworm larvae at random occasions. 

One treatment group (Group 3) was challenged with infectious L3 larvae only once (at week 2; two weeks after M-S-P), while the other treatment group (Group 4) was challenged twice (at week 2 and week 7; two and three weeks after M-S-P, respectively).

### 2.2. Dosage Calculation 

Rats received sarolaner/moxidectin/pyrantel (half of the Simparica Trio™ tablet, 10.1–20 kg size; Zoetis, Sydney, New South Wales, Australia). This tablet contains 24 mg sarolaner, 480 µg of moxidectin and 100 mg pyrantel [16]. Therefore, each rat voluntarily consumed or was gavaged with approximately 240 µg of moxidectin. For a rat weighing 500 g, this equates to a dose of 480 µg/kg. The dose calculation assumes that the tablet is homogenous in its formulation, which is unproven. 

### 2.3. Examination for Patent Infection and Presence of A. cantonensis L1 in Rat Faeces

The Baermann technique was used to extract *A. cantonensis* L1 in rat faeces [17,18]. A wet preparation slide with a coverslip was made and viewed using conventional light microscopy using the 10× objective lens. Larvae in faeces were highly motile when detected.

### 2.4. Necropsy Examination of Rats to Detect Adults A. cantonensis in the Right Ventricle and Pulmonary Artery of Rats

All rats were humanely euthanised at week 14 by inspiration of 100% carbon dioxide. Each rat was weighed using electronic scales immediately after euthanasia. The heart and lungs were removed from the chest cavity by a combination of blunt and sharp dissection to locate all adult nematodes in the right ventricle and/or pulmonary arteries. The numbers of male and female worms were determined by examination under a dissecting microscope. The worms were small, ranging from approximately 15–25 mm in length with the females having a slightly larger diameter and longer length, as well as the ‘barber’s pole’ appearance, caused by the alimentary tract (containing digested blood) and reproductive tract being wrapped around each other [2,4,9]. 

### 2.5. Statistical Analysis

The weights of treated and control rats were compared using the Mann–Whitney U test. A two-tailed Fisher’s Exact test was used to compare the number of adult *A. cantonensis* worms present in treated versus control rats. The two M-S-P treatment groups (of 5 rats) were combined (10 rats in total) and compared to the 8 control rats.

## 3. Results

There were no mortalities or treatment-related adverse reactions over the course of the study. All rats continued to grow and increase in body mass during the experiment. None of the rats appeared dyspneic (at rest) at any time during the experiment.

Both control groups, Group 1 and Group 2, were successfully infected with *A. cantonensis* L3 larvae (Table 2). A total of seven out of eight control rats had adult worms (11–23) present on necropsy at Week 14 and all seven were positive for *A. cantonensis* L1 larvae on Baermann examination of faecal pellets. One of the eight control rats (from Group 1) had no *A. cantonensis* L1 larvae in its faeces and only a single *A. cantonensis* adult male worm present in a pulmonary artery. Considering that approx. 30 L3 were given to each rat, the resulting worm burden (median 14.5 worms; IQR 11.5 to 16.5 worms) per rat was consistent with about half of the larvae reaching maturity. Noticeable lesions in the pulmonary parenchyma were present in all the control rats except for the rat with the single worm infection (Figure 1). The gross pulmonary lesions were mostly localised in the caudodorsal portions of the lungs.

All four Group 3 rats that were treated with M-S-P on weeks 0, 4 and 8 weeks remained negative despite a single challenge with *A. cantonensis* L3 larvae at week 2 (Table 1). Group 4 rats were treated on weeks 0, 4 and 8 and challenged with *A. cantonensis* L3 larvae at week 2 (2 weeks after M-S-P) and again at week 7 (3 weeks after M-S-P; Table 1). All four rats were negative for *A. cantonensis* L1 larvae in their faeces, but at necropsy, two out of five rats each had a single female *A. cantonensis* recovered from the pulmonary arteries. 

Thus, 8/10 rats dosed with M-S-P had zero lungworms at necropsy; in the remaining two rats, which might not have received a full dose of M-S-P for technical reasons, a single female worm was detected. In contrast, 8/8 control rats were infected with *A. cantonensis*, with a median of 14.5 worms per rat detected at necropsy and patent infections with motile L1 abundant in faecal pellets. The difference in infection rates was highly significant (two tailed Fishers Exact; *p* = 0.0011; [19]).

We can also express the results in terms of percentage burden reduction. In control rats, 109 worms were present in 8 rats; whereas in rats given M-S-P monthly, only 2 worms were present in 10 rats; so, the burden reduction was 13.625 worms per rat to 0.2 worms per rat, which is a 98.5% reduction. Monthly M-S-P administration prevents the shedding of L1 larvae in all rats, regardless of whether there was one or two challenges with infective larvae. Likewise, monthly M-S-P prevented the development any of discernible gross lung pathology at necropsy examination.

There was no significant difference between the weights of the treated and control rats at the end of the experiment (Mann–Whitney U test; U = 22; *p* = 0.12; [20]).

## 4. Discussion

M-S-P is a fixed dose combination of three anthelmintic drugs designed to be administered monthly in dogs to prevent heartworm disease, tick paralysis, flea, lice and mite infestations, and intestinal nematode infections [16]. Its spectrum covers all important and common helminth infections except tapeworms (cestodes). The experiments described here represent a model (pilot experiments) for a conceptually similar study that we hope to undertake in dogs to determine if it is possible to prevent them from getting neuroangiostrongyliasis when given M-S-P as a monthly preventative. 

In this study, rats were used as a surrogate for dogs because it is much easier to obtain animal ethics approval for rat experiments in our jurisdiction and such trials are substantially less expensive. Our aim was to use the moxidectin component of M-S-P to interrupt the life cycle by killing migrating L3 larvae in the CNS, thereby preventing them maturing into adult nematodes and reproducing within the definitive host [3,9]. Because M-S-P is given either 1 week or 2 weeks after larval challenge, the L3 larvae have all left the gut and entered the CNS. Furthermore, because the L3 are no longer in the gut, the pyrantel in the M-P-S does not contribute any effect on larvae, as pyrantel does not achieve effective concentrations in the CNS, which is where the L3 larvae are at this point in time. Finally, sarolaner is an isoxazoline ectoparasiticide thought to have no effect on nematode larvae, but is rather a selective inhibitor of arthropod γ-aminobutyric acid- and l-glutamate-gated chloride channels in fleas and ticks.

What we would actually like to show in dogs is that moxidectin concentrations in plasma are sufficiently high to kill L3 before they enter the CNS, but this is harder to prove in rats, as the migration of modest larval burdens do not usually cause observable neurologic signs. In rats, the L3 larvae can be found circulating in blood within a few hours of inoculation, and within 24 h, L3 larvae have entered the spinal cord and brain where they grow, moult twice, approximately on days 7 and 14 post-infection, then reach the subarachnoid space from where they eventually leave the CNS as L5 larvae, on the way to the right ventricle and pulmonary arteries [3,4,21].

The pharmacokinetics of moxidectin given orally to Wistar rats had not been determined at the start of the research project, which commenced in mid-February 2021 during the COVID pandemic. It was presumed that moxidectin would have a half-life in rats comparable to what has been reported in the dog, such that monthly administration would have substantial and cumulative activity, such that when rats would be challenged with an oral dose of infective larvae, blood concentrations of moxidectin might be sufficient to kill the larvae before they migrate to the CNS. Based on this notion, the experimental protocol set out in Table 1 was constructed.

However, with the benefit of the data from Buchter and colleagues published in July 2021, serum levels of moxidectin are essentially nil by 48 h after oral administration of 500–750 µg/kg [22]. The half-life of moxidectin given orally to dogs is very long viz. 621 h (13.9 to 25.9 days), while in Wistar rats the half-life is only 10.4 h; this means that therapeutic blood levels of moxidectin are unlikely to be maintained for more than 2–3 days in our rats, whereas in dogs the coverage extends to approximately a full calendar month [14,15,22]. In other words, if M-S-P is given at t = 0, 4 weeks and 8 weeks, then blood concentrations would have fallen to zero by the time the inoculum of 30 L3 larvae were given at week 2 or week 7. 

Therefore, the action of moxidectin in M-S-P is attributable to its larvicidal action on migrating L3, L4 and L5 larvae in the CNS when it is given at 4 weeks and 8 weeks, with the M-S-P reaching short-lived therapeutic levels 2 weeks or 1 week, respectively, after larval challenge (Figure 2). Thus, our results confirm those of Schmahl et al. [18] who showed that moxidectin given transdermally at 4–32 mg/kg (with imidacloprid as the topical Advocate™; Elanco) at 15 days post-infection was highly effective at killing the ‘CNS-dwelling larvae’ of *A. cantonensis* [3,4,18,21]. It is, in some respects, remarkable that the death and disintegration in the order of 11–23 larvae (expected to be approx. 3 mm long [21]; Figure 2) did not produce more discernible neurological signs that might be appreciated even by a short daily examination. 

In our study, M-S-P was administered either 7 or 14 days ***after*** challenge with infective larvae. Larvae are in the CNS at this stage and remain susceptible to moxidectin, as this macrocyclic lactone readily crosses the blood brain barrier (BBB) [23,24]. Furthermore, inflammation from the migrating larvae may have caused the BBB to become leaky and pro-inflammatory cytokines are known to inhibit the p-glycoprotein pump, allowing more of the lipophilic drug to enter the extracellular fluid around the parasite [23,24]. This means that in the first rat treatment group (Group 3), it was the M-S-P containing moxidectin given 2 weeks after larval challenge which killed larvae migrating through the CNS, when the moxidectin blood and CNS concentrations were sufficiently high. The same was true in the first challenge of the second treatment group of rats (Group 4), but in the second challenge, the M-S-P containing moxidectin given 1 week after larval challenge was the dose that was effective. It was fortuitous that the dose of moxidectin we selected based on the older literature and allometric scaling was similar to doses informed by recent pharmacokinetic studies. The chosen dose of approx. 480 µg/kg was close to the most efficacious dose (500 µg/kg) used for treating *Strongyloides ratti* infections in rats [22].

As stated, this study was conducted in the spirit of being a pilot experiment, as there had been limited previous research into the chemoprophylaxis of *A. cantonensis* infection in rats or dogs. The product M-S-P was selected as it represents the drug combination of greatest potential to prevent the important parasitic diseases of companion dogs in eastern Australia, including rat lungworm disease. The ‘palatable chew’ formulation of M-S-P, however, contributed to dosage inaccuracy, in that we do not know whether the active was uniformly distributed in the tablet, nor were we sure that rats swallowed all of the drug when it was given by gavage under light gaseous anaesthesia. This may have contributed to the two rats with a single worm infection despite monthly M-S-P administration. Although rats would eat the tablet once, they would not eat them subsequently, probably because of the development of a food aversion to one or more of the active ingredients. It was thus necessary to create a drug suspension administered by gastric lavage under anaesthesia for subsequent administrations. The half a tablet dose was not fully administered, as a small amount of residual solution was always left in the dead space of the pipette. It would have been much easier if the rats could have consumed the complete dose on their own. Presumably, this would not be a problem if the experiments were repeated in dogs, the species for which the palatable chew was developed.

A further complication of the oral gavage technique also applies to administering the infective L3 larvae. The exact number of L3 larvae administered orally to each rat was variable from rat to rat, which would in part explain some of the variation in worm numbers observed in control rats at necropsy. When performing oral gavage, a portion of the solution will always remain inside the dead space of the pipette. Thus, a few L3 larvae probably remained in the pipette after each administration. For this reason, a new pipette was used for each gavage. In relation to the single control rat that had a much lower worm burden than the rest of the controls, it is suspected that the oral gavage of the L3 larvae was not technically perfect in this individual. This control rat did develop a single worm infection, but the worm burden was smaller than the other controls, suggesting most larvae were not swallowed, or were swallowed and then regurgitated. Indeed, a risk of oral gavage under gaseous anaesthesia is that gastric reflux and aspiration pneumonia can occur with poor technique and especially with insufficiently deep anaesthesia [25].

Moxidectin is an anthelmintic used to prevent other nematode species such as the heartworm *Dirofilaria immitis* [12] and in the UK and Europe, *Angiostrongylus vasorum* [11,13,26]. This latter parasite causes ‘French heartworm’, a complicated disease resulting from the presence of many adult worms in the pulmonary arteries of infected dogs. This canine disease can be prevented by the monthly administration of moxidectin, which kills both migrating larvae and adult worms in the pulmonary arteries. The efficacy of this drug is in part due to its very long half-life in the dog, which results in cumulative kinetics with effective blood concentrations of moxidectin present for the entire month when the product is given every four weeks. Moxidectin targets nematodes by opening chloride channels in their cell membranes, causing lethal paralysis [27]. Based on the results presented here and in the work of Schmahl and colleagues [10], the moxidectin in M-S-P provided blood and CNS concentrations sufficient to prevent rat lungworm developing to patency following an oral challenge with L3 larvae. Interestingly, Schmahl et al. [10] observed that the doses of moxidectin they used had no effect on adult lungworms in the pulmonary artery of rats, but given the recent work showing the short half-life of moxidectin in the rat [12], it seems most likely that mature worms require more than 1–2 days of effective moxidectin blood concentrations to succumb. 

Moxidectin is generally considered superior to other macrocyclic lactone anthelmintics such as ivermectin, selamectin and milbemycin, with moxidectin showing a faster onset of activity and a greater efficacy, for example against L3 larvae of *Strongyloides* spp. [22]. Moxidectin also has a substantially longer half-life and a greater area under the curve when given orally to dogs compared to when it is given to rats [14,15,16,22]. Moxidectin remains in the blood for sufficiently long in the dog that the drug accumulates when given monthly, resulting in progressively higher serum and tissue concentrations. Such prolonged high levels likely explain its efficacy against mature *A. vasorum* worms in the pulmonary arteries of dogs [14,15]. 

In the dog, the prolonged kinetics of moxidectin would likely prevent neuroangiostrongyliasis by killing infective L3 larvae before they would reach the CNS, a mechanism not observed in these experimental rats because of the much shorter half-life of moxidectin in this species. However, this needs to be confirmed experimentally. Moxidectin in various formulations is already being used prophylactically with this intention in highly endemic areas, such as in Hawaii, Sydney, Brisbane and along the east coast of Australia. A further issue relates to whether the long-acting depot formulations of moxidectin (Proheart SR12™; Zoetis) would produce sufficiently high concentration in serum to kill infective third stage larvae of *A. cantonensis*, and for how long [14].

Completion of this study has helped to further expand the knowledge of anthelmintics used in rats. Monthly moxidectin would prevent pet rats from developing a patent *A. cantonensis* infection should they eat an infected slug or snail.

## 5. Conclusions 

This study demonstrates unequivocally that administering Simparica Trio™ containing moxidectin can interrupt the lifecycle of *A. cantonensis* in rats by causing lethal paralysis of infective larvae migrating through the spinal cord, peripheral nerves, brain and subarachnoid space. This research conducted on rats can be used to help guide preventive care in domestic animals such as dogs, as well as wildlife and zoo animals in high prevalence areas with endemic *A. cantonensis*. 

## Figures and Tables

**Figure 1 pathogens-12-00305-f001:**
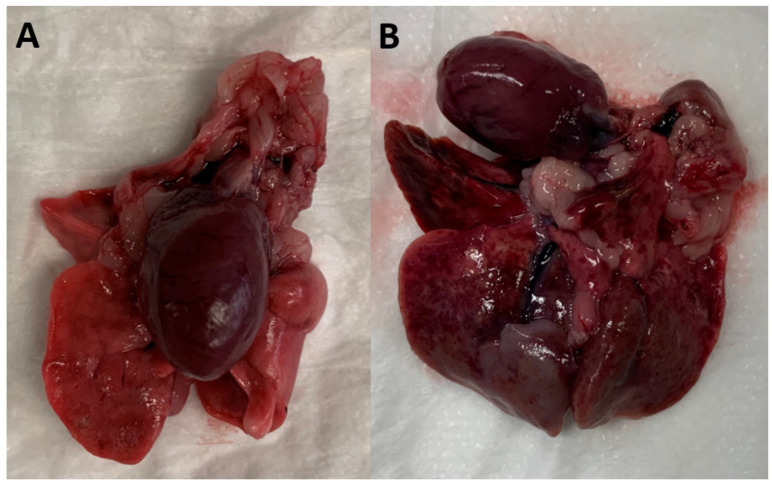
Heart and lungs dissected from Wistar rats at the end of the experiment ex vivo. In (**A**), a rat treated with moxidectin in M-S-P has normal heart and lungs, while a control rat with a moderate burden of mature *A. cantonensis* is shown in (**B**) on the right.

**Figure 2 pathogens-12-00305-f002:**
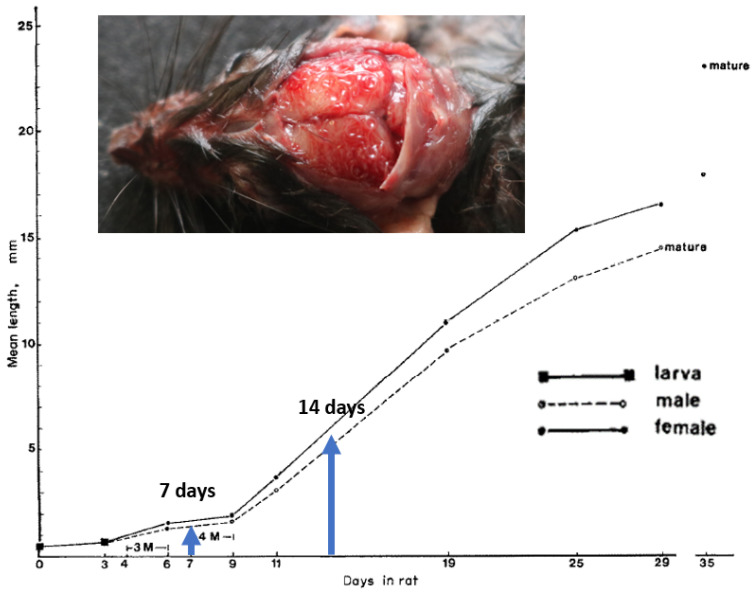
Growth of *A. cantonensis* in experimental white Wistar rats; 3 M = 3rd moult; 4 M = 4th moult. The blue arrows indicate the size of the migrating larvae at 7 days and 14 days when they would be exposed to moxidectin following the administration of Simparica Trio™ by ingestion following gastric lavage. Inset: photo of a wild rat after removal of the calvarium to demonstrate young adult worms (L5) in the subarachnoid space overlying the cerebrum. Photograph courtesy of Lydia Tong and Derek Spielman, Taronga Zoo; original diagram adapted from Manoon Bhaibulaya’s classic paper in 1973 [25], with modifications.

**Table 1 pathogens-12-00305-t001:** Timetable that outlines the schedule of when rats were challenged with *A. cantonensis* L_3_ larvae and when the Simparica Trio™ (M-S-P) dose was administered.

Week	Group 1(Control; *n* = 4)	Group 2(Control; *n* = 4)	Group 3(ST; *n* = 5)	Group 4(ST; *n* = 5)	
0			**M-S-P**	**M-S-P**	
1					
2	**30 L_3_ larvae PO**		**30 L_3_ larvae PO**	**30 L_3_ larvae PO**	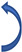
3				
4			**M-S-P**	**M-S-P**
5					
6					
7		**30 L_3_ larvae PO**		**30 L_3_ larvae PO**	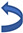
8			**M-S-P**	**M-S-P**
9					
10					
11					
12					
13					
14	Necropsy	Necropsy	Necropsy	Necropsy	

Orange font highlights infective dose of L3 larvae; Blue highlights dosing with moxidectin in Simparica Trio; the blue arrows indicate the infective larvae that are targeted by the moxidectin in M-S-P given 14 or 7 days later.

**Table 2 pathogens-12-00305-t002:** Pooled results of the worm burden found in the control and treatment groups. Rats were dosed with approx. 30 L_3_ larvae.

Parasite Status	Group 1 (Control; *n* =4) One Challenge at 2 Weeks	Group 2 (Control; *n* = 4) One Challenge at 7 Weeks	Group 3 (M-S-P; *n* = 5) One Challenge at 2 Weeks	Group 4 (M-S-P; *n* = 5) Two Challenges at 2 and 7 Weeks
**Infected**	4/4	4/4	0/5	2/5
** *A.c.* ** **Male**	4,5,10,1	17,8,10,6	0,0,0,0,0	0,0,0,0,0
** *A.c.* ** **Female**	8,6,6,0	6,9,6,7	0,0,0,0,0	1,0,0,0,1
** *A.c.* ** **Total**	12,11,16,1	23,17,16,13	0,0,0,0,0	1,0,0,0,1
**L_1_ larvae in** **fresh faeces**	3/4 rats positive (1 rat with a single male worm was negative)	4/4 positive	0/5 positive	0/5 positive (2/5 rats each had a single female worm and were negative)
**Lung** **lesions**	3/4	4/4	0/5	0/5
**Body weight** **(g)**	690,674,662,595 (median 668)	580,673,595,591 (median 593)	646,586,595,616,538 (median 595)	584,565,633,616,589 (median 589)

***A.c.*** *Angiostrongylus cantonensis.*

## Data Availability

All the data applicable to this investigation are presented in this manuscript.

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
