# Peer review of "Successful Removal of Angiostrongylus cantonensis Larvae from the Central Nervous System of Rats 7- and 14-Days Post-Infection Using a Product Containing Moxidectin, Sarolaner and Pyrantel Embonate (Simparica Trio™) in Experimental Infections"

_pathogens, 2023, doi:10.3390/pathogens12020305_

Round 1

Reviewer 1 Report

Manuscript Number: pathogens-2127374; Henry et al., Successful removal of Angiostrongylus cantonensis larvae from the central nervous system of rats 7- and 14-days post-infection using a product containing moxidectin in experimental infections

The manuscript is written very clearly and all aspects are described in detail. Thus it is easy to understand. However, some issues need to be addressed.

The biggest problem of the MS is that anthelmintic efficacy against migrating larvae is attributed only to moxidectin. In fact, a combination of three different active compounds (sarolaner+moxidectin+pyrantel embonate) in chewable tablet was used in this experiment. The killing effect on A. cantonensis migrating larvae could be caused by each compound separately (in deed, sarolaner is less probable) but also by the effect of their interaction. In addition, recent studies showed that Pyrantel pamoate is effective drug against rat lungworm larvae (see Jacob et al. 2021, DOI: 10.1017/S0031182020001146, Jacob et al. 2022 https://www.sciencedirect.com/science/article/pii/S2211320722000069).

Please, modify in the title and entire text “moxidectin” to “moxidectin+sarolaner+pyrantel embonate”. Please, give a comment on which compound is responsible for anthelmintic effect to Discussion as well. And rewrite corresponding parts in Discussion (255-263, 336-356)

It is not correct to use commercial name (Symparica Trio) and its abbreviation (ST) in the entire text. Names of active substances are preferred. Use the name of commercial drug only in M&M (origin of drug Line 140-141). In other parts of the text, authors should use “sarolaner+moxidectin+pyrantel pamoate” or in abbreviated form. 

Lines 82-91: well, that is true that MXD is a drug with longer halftime. However, there are very limited data that MXD is effective against A. cantonesis (only Schmahl). I think this should be emphasized. We currently do not know which compound is the best choice for A. cantonensis. For example, Jacob et al. 2021 (DOI: 10.1017/S0031182020001146) suggests that Albendazole, Pyrantel and other anthelmintics acts better than avermectins against A. cantonesis L3. Please modify this paragraph.

Lines 113-114: How the dissolved tablet in distilled water looked like? Was it clear solution or suspension? Please, be aware that this action might also influence absorption and bioavailability of active substances, especially in case of strongly lipophilic moxidectin.

Lines 255-264: Again, as stated above. How do authors know that MXD was responsible for the therapeutic effect? Why not pyrantel? Or their interaction? Please modify the whole paragraph.

Reviewer 2 Report

General comments: In this manuscript titled “Successful removal of Angiostrongylus cantonensis larvae from the central nervous system of rats 7- and 14-days post-infection using a product containing moxidectin in experimental infections”, it was aimed to determine if monthly moxidectin administration orally could prevent symptomatic neuroangiostrongyliasis in dogs after A. cantonensis infection. This study demonstrates unequivocally that administering Simparica Trio containing moxidectin can interrupt the lifecycle of A. cantonensis in rats by causing lethal paralysis of infective larvae migrating through the spinal cord, peripheral nerves, brain and subarachnoid space. This research conducted on rats can be used to help guide preventive care in domestic animals such as dogs as well as wildlife and zoo animals in high prevalence areas with endemic A. cantonensis. 

In general the paper is well done,the age and weight of animals are very well controlled as well as the methodology is well explained and provides sufficient detail to understand the research work. However, there are few type mistakes and is easy to follow.

1.     In my opinion, the abstract section can be more brevity. The change of methods of administration no need to be describe in detail.

2.     Based on experience, rats can give intragastric administration directly. Why do you administer to rats via orogastric lavage under light isoflurane anesthesia? If appropriate, please try to explain whether the narcotic drugs can impact ST absorption or not.

3.     It’s better to clarify the effect of sarolaner and pyrantel (combination of sarolaner/moxidectin/pyrantel embonate) on the treatment of A. cantonensis.

4.     References format should be unified(Reference 1, 9, 27).
